# "You know that we travel a lot": Mobility narratives among female sex workers living with HIV in Tanzania and the Dominican Republic

Zoé M. Hendrickson[1,2]*, Maria De Jesus[3], Clare Barrington[4], S. Wilson Cole[1], Caitlin E. Kennedy[5], Laura Nicole Sisson[1], Mudia Uzzi[1,6], Yeycy Donastorg[7], Martha Perez[7], Hoisex Gomez[7], Jessie Mbwambo[8], Samuel Likindikoki[8], Deanna L. Kerrigan[9]

1 Department of Health, Behavior and Society, Johns Hopkins Bloomberg School of Public Health, Baltimore, Maryland, United States of America, 2 Department of Behavioral and Community Health Sciences, University of Pittsburgh School of Public Health, Pittsburgh, Pennsylvania, United States of America, 3 School of International Service, American University, Washington, DC, United States of America, 4 Department of Health Behavior, University of North Carolina at Chapel Hill Gillings School of Global Public Health, Chapel Hill, North Carolina, United States of America, 5 Department of International Health, Johns Hopkins Bloomberg School of Public Health, Baltimore, Maryland, United States of America, 6 Department of Mental Health, Johns Hopkins Bloomberg School of Public Health, Baltimore, Maryland, United States of America, 7 Instituto Dermatologico y Cirugia de la Piel, Santo Domingo, Dominican Republic, 8 Muhimbili University of Health and Allied Sciences, Dar es Salaam, Tanzania, 9 Prevention and Community Health Department, Milken Institute School of Public Health, the George Washington University, Washington, DC, United States of America

* zhendri1@jhu.edu

**Data Availability Statement:** Data cannot be shared publicly because the study involves sensitive data collected from a small, vulnerable

## Abstract

Female sex workers (FSW) are highly mobile, which may result in reduced access to and use of health services and increased risk for poor health outcomes, particularly for those living with HIV. Mobility includes spatial, temporal, and social elements that are not fully captured by quantitative measures. We conducted two rounds of in-depth interviews with FSW living with HIV in Iringa, Tanzania (n = 20), and Santo Domingo, Dominican Republic (n = 20), to describe mobility experiences and compare mobility narratives across settings. We integrated a thematic analysis of all interviews with a narrative analysis of a subset of 10 information-rich interviews (five in each country) with women who had recently traveled, for sex work or another reason, outside of their hometown. Across narratives, FSW living with HIV traveled locally or to seasonal destinations, for short and long periods. Social factors influencing mobility included economic drivers; risk of arrest, harassment, or violence; anonymity and/or familiarity; social relationships; and clients' mobility. Spatial, temporal, and social factors intersected in unique ways in FSW's mobility experiences, yet distinct mobility typologies were evident across settings and destinations. Together, mobility narratives of FSW living with HIV can inform quantitative research on mobility typologies in Tanzania, the Dominican Republic, and elsewhere. With the potential for economic circumstances, climate change, and other emergencies to increase people's mobility around the world, researchers and practitioners can learn from the lived experiences of FSW to inform whether and how to

population. This study focuses on a small, marginalized population group, with data including sensitive information related to their HIV status as well as engagement in sex work. The sensitivity of the information, as well as the potential risk posed to participants, is a significant ethical concern that prevents the sharing of the de-identified data. Please contact the Johns Hopkins Bloomberg School of Public Health institutional review board at JHSPH.irboffice@jhu.edu.

**Funding:** This research was funded in part by a 2019 developmental grant from the Johns Hopkins University Center for AIDS Research, an NIH funded program (1P30AI094189), which is supported by the following NIH Co-Funding and Participating Institutes and Centers: NIAID, NCI, NICHD, NHLBI, NIDA, NIMH, NIA, FIC, NIGMS, NIDDK, and OAR. The content is solely the responsibility of the authors and does not necessarily represent the official views of the NIH. This research was supported by the 2019 Gustav J. Martin award for innovative HIV research from the Johns Hopkins Bloomberg School of Public Health. The data analyzed as part of this publication was collected as part of a study funded by the US National Institute of Mental Health through R01 MH110158. It was facilitated by the infrastructure and resources of the Johns Hopkins University Center for AIDS Research, a US NIH-funded program (1P30AI094189). It was also supported by the following NIH Institutes and Centers: NIAID, NCI, NICHD, NHLBI, NIDA, NIMH, NIA, FIC, NIGMS, NIDDK and OAR. Laura Sisson is supported on a NIMH T32 training grant (T32MH122357; PI: Stuart).

**Competing interests:** The authors have declared that no competing interests exist.

tailor and improve the accessibility of HIV care and treatment interventions based on spatial, temporal, and social characteristics of mobility.

# 1. Introduction

Female sex workers (FSW) globally are highly mobile [1–5]. Research from Sub-Saharan Africa, Latin America, and elsewhere has shown that mobility can reduce FSW's access to and use of health services, increase substance use and HIV risk, and lead to poor health outcomes [4, 6–12]. For FSW living with HIV, mobility can prevent access to HIV care and treatment services, leading to interruptions in care [13–15]. Mobility can also affect participation in community-based HIV interventions, with studies suggesting that mobility restricts FSW's interactions with peer facilitators [16] and limits engagement in HIV prevention and care interventions [14, 17, 18]. However, a study in Zimbabwe, consistent with findings from a recent systematic review, demonstrated how associations vary with context, citing mixed evidence on relationships between mobility and antiretroviral therapy (ART) adherence and viral load and calling for improved measurement of FSW's mobility [15, 19]. Understanding the nature of mobility and variations across settings is therefore essential to designing better programs and services for FSW living with HIV [20].

## 1.1. Conceptualizing FSW's mobility

Mobility is a process [21] that includes spatial, temporal, and social elements. As Martinez-Donate et al. (2015) and Zhang et al. (2017) described in their exploration of HIV risk, mobility spans places and times from pre-departure to transit, destination, and return [22, 23]. Indeed, mobility is shaped by individuals' trans(spatial) social fields [24] that are multiscalar–constituted by relationships and unequal power dynamics across space and time [25, 26]. People therefore intersect with places that are the product of multi-level networks of individuals and institutions [25].

Recently, Thorp et al. (2023) proposed a research agenda on mobility and HIV care engagement, highlighting the need to understand what aspects of mobility are most influential in people's access to and utilization of healthcare services [27]. To do this, it is essential that we have a robust understanding of the multiscalar elements that characterize people's mobility. Most quantitative measures of FSW's mobility focus on work-related mobility broadly, with FSW mobility defined by any travel, duration of travel, or destination of travel for sex work within a given period [2, 11, 12, 28]. These measures often do not identify the many characteristics–spatial, temporal, and social–of mobility. Spatial characteristics of mobility include, for example, destination(s) of travel, while temporal characteristics can include duration or frequency of travel. Social characteristics include reasons for mobility, living or work environments during travel, or social relationships and interactions during mobility [1, 19]. Mobility may be linked to FSW's work, such as travel to meet new clients [9, 29], or a response to the socio-structural context in which FSW live and work [11, 28, 30–32]. Mobility can also disrupt FSW's social relationships, leading to mistrust [33] and fragmentation of the FSW community [16, 34], including absences that reduce access to social support during transit and upon return.

Recent research has quantitatively identified distinct typologies of mobility among FSW [1]. Davey et al. (2019) collected information from FSW in Zimbabwe who had worked in sex work in another location within the last year to capture where they had been, duration of recent trips, reasons for mobility, and use of healthcare services while traveling. Using latent

class analysis, they identified five typologies that included infrequent work-related mobility; frequent, domestic work-related mobility; frequent, international work-related mobility; infrequent non-work-related mobility; and client-influenced mobility. However, the nuances of these typologies are not fully captured or contextualized by standard quantitative measures. Indeed, the authors highlighted that qualitative data could have aided in understanding these typologies in greater depth. Here, we build on Davey et al. (2019)'s work in Zimbabwe to explore FSW's mobility experiences in two countries: Tanzania and the Dominican Republic (DR). We draw on in-depth qualitative interviews to: (1) describe mobility experiences among FSW living with HIV in Tanzania and the DR and (2) compare mobility narratives within and across settings. We aim to inform future contextualized research on FSW's mobility and HIV care and treatment interventions.

## 2. Materials and methods

This section follows recommendations from the Standards for Reporting Qualitative Research (SRQR) for the reporting of details about the study's materials and methods [35].

### 2.1. Study context and design

This study drew on two rounds of in-depth interviews conducted in late 2018 and late 2019/ early 2020 with FSW living with HIV in Iringa, Tanzania, and Santo Domingo, DR. Participants in in-depth interviews were recruited from a larger longitudinal observational study, entitled *Stigma, cohesion and HIV outcomes among vulnerable women across epidemic settings* [36]. Participants were recruited for qualitative interviews between October 24, 2018, and February 25, 2020.

The DR and Tanzania differ substantially in ways relevant to FSW's mobility. Santo Domingo is the largest city and capital of the DR, with a population of nearly 1 million. The city is a seaport and economic hub. Industries, manufacturing, and tourism are essential to the local economy [37]. Tourism influences internal mobility and migration among people in the DR, including FSW [10, 38]. Iringa is a region in the Southern Highlands of Tanzania. The administrative capital is Iringa town, with a population of less than 200,000. The Tanzanian-Zambian (Tan-Zam) highway winds through Iringa, and agricultural production, movement, and transport are features of the local economy [39–41].

### 2.2. Participants and data collection

FSW were eligible for the parent study if they were cisgender women, at least 18 years of age, living with HIV (confirmed HIV-positive diagnosis) and exchanged sex for money within the past month [42]. The parent study drew on existing cohorts in Tanzania and the DR [36]. A subset of 20 FSW participating in these cohorts in each country was selected to participate in in-depth interviews. Interviews were conducted in Swahili (in Tanzania) or Spanish (in the DR) by local, trained interviewers with extensive experience in qualitative research. Individuals were recruited purposively based on viral load, with half virally suppressed (<400 copies/mL) and half not virally suppressed (≥400 copies/mL) [42]. Viral load was used as part of the sampling approach to address complementary study aims exploring the complex dynamics surrounding FSW's differential experiences with HIV services and outcomes. In-depth interviews explored social relationships, work environments, HIV care and treatment, healthcare system interactions, and health and well-being. In the first round of interviews, mobility emerged organically as an important theme. In the second round of interviews, we included a mobility module to explore recent mobility experiences–including destination and frequency of

mobility, reason for travel, experiences in transit, and social relationships at one's place of origin and destination.

We reviewed participants' responses during qualitative interviews as well as responses to mobility-related questions in the three rounds of quantitative surveys conducted by the parent study. Among the 40 FSW interviewed qualitatively, 15 out of 20 in the DR and 12 out of 20 in Tanzania described recent mobility, generally or for sex work, either in a qualitative interview or in at least one of the three rounds of quantitative surveys conducted by the parent study. These 27 FSW were included in the first phase of analysis described below. Those FSW excluded from analysis at this stage did not describe any recent mobility either in the qualitative interviews or in a quantitative survey. In the second phase of analysis, we then focused our analysis on a sub-sample of these 27 FSW who had participated in both rounds of qualitative data collection to ensure that they were asked questions about mobility. Among those participating in both qualitative interviews, five FSW in the DR and five FSW in Tanzania discussed mobility for sex work or general mobility during the second qualitative interview. Thus, in the second phase of analysis, we narrowed our analytical focus to this sub-sample of information-rich interviews with 10 FSW living with HIV.

## 2.3. Analytical approach

Our qualitative data analysis approach was informed by Maxwell & Miller (2008)'s framework for how to integrate categorizing and connecting strategies in qualitative data analysis [43, 44]. We used categorizing strategies (i.e., coding and thematic analysis) to sort and group data into categories to identify similarities and differences within and across participants' stories. We then used connecting strategies (i.e., analytical memos and narrative analysis) to retain the context of participants' stories, focusing on the connections and relationships that exist within these stories [43, 44]. We then compared mobility experiences and explored how themes connected in FSW's narratives, moving from categorizing to connecting to categorizing in a multi-phased approach [43, 44].

In the first phase of analysis, we followed an iterative coding process–using deductive and inductive codes–for the first round of qualitative transcripts. We conducted a thematic analysis to identify shared and divergent emergent themes about mobility in the DR and Tanzania, which informed the mobility module developed for the second round of qualitative interviews conducted in late 2019 and early 2020. We implemented a second round of coding with the full set of qualitative transcripts following the second round of interviews. We linked transcripts from round 1 and round 2 interviews for each participant, coding them together, to encourage a holistic coding and analysis process. We adapted the codebook from round 1 by adding *a priori* codes, guided by topics in the mobility module, and emergent codes.

In the second phase of analysis, we used a narrative analytical approach to interrogate a sub-sample of transcripts from interviews with 10 FSW, as detailed above, to describe FSW's mobility narratives in the DR and Tanzania. This approach examined participants' intact stories [45]. We used an iterative analytical memo-ing process to analyze interviews, which used connecting and categorizing approaches [42–44]. The memos were structured to categorize information about FSW's mobility: the *what* (how mobility was described or defined, what happened, and health system interactions during travel), *when* (duration and frequency), *why* (reason for travel), *where* (destination), *how* (the process, including preparations and transit), and *who* (social relationships during travel). Memos included connecting summaries of participants' lived realities over the two interviews, salient quotations, and space for researcher reflexivity [46, 47]. We used memos to develop the mobility narratives included in this article.

Visualizations of participants' destinations mentioned, created using arcGIS, complemented the written memos and mobility narratives.

One research team member drafted memos from each country (ZMH and MDJ), with another critically reviewing the memo. Consensus meetings were held iteratively during analysis. ZMH and MDJ compared emergent themes across settings to highlight the co-occurrence of spatial, temporal, and social factors influencing mobility experiences in Tanzania and the DR. This process identified cross-cutting factors including characteristics of FSW mobility, motivations for travel, and social relationships. We present here results of this multi-phased analysis, featuring mobility narratives and drawing on examples from the subsample of 10 FSW to highlight similarities and differences. All names are pseudonyms.

### 2.4. Ethics statement

This study was approved by the Johns Hopkins Bloomberg School of Public Health (JHSPH) institutional review boards in the United States (Ref #: 00007065), the Muhimbili University of Health and Applied Sciences (MUHAS) IRB and National Institute of Medical Research (NIMR) IRB in Tanzania (Ref #: 1593), and the Instituto Dermatológico y Cirugía de la Piel (IDCP) in the DR (Ref #: 7065). Participants provided oral informed consent prior to data collection. Building on previous experience conducting HIV-related research with people who engage in sex work, due to the sensitive topics addressed in this study, and since the study included participants with historically marginalized identities, the study utilized oral consent procedures to protect participants' confidentiality.

Additional information regarding the ethical, cultural, and scientific considerations specific to inclusivity in global research is included in the Supporting Information (S1 Checklist).

## 3. Results

First, we spotlight two mobility narratives of FSW in Tanzania and the DR. These participants' stories varied by destination and frequency of travel, motivations for travel, and social relationships. Additional narratives are included as supplementary material (S1 Table).

### 3.1 Select mobility narratives from FSW in Tanzania and the DR

**3.1.1. Anisa, who travels seasonally and in response to risk.** Anisa, a Tanzanian woman, was "used to traveling frequently." She was new to the Iringa region, arriving in Ilula a few years ago. She traveled to visit family and for work. She described recent trips to Mtera and visits to Dodoma and mining areas in Mbeya or Chunya during the dry season. She often spent time in Dodoma and was familiar with that environment, traveling there by car. She frequently stayed in guesthouses with other women when traveling. Once she stayed in a dormitory in Mbeya but opted to change to a guesthouse with a bit more privacy. Her trips were longer, sometimes lasting a month or longer. When traveling for sex work, she moved on if she did not get enough work or money in one location. The relatively high cost of living in Iringa played an important role. Seasons also played a role. Rainfall had recently prevented her from going back to Dodoma. She traveled with or for clients and moved to smaller villages because of police or negative interactions with fellow workers. A fellow worker in Dodoma, for instance, grew jealous of her income and spoke ill of her, taking her HIV medication to show others. In response, Anisa moved to a new location on the other side of Iringa. Friends and family were sources of social support. Her brother supported her financially during travel, and friends shared medications or gave advice on when and where to travel. Anisa had people with whom she would travel for work, sometimes as far as Mozambique.

**3.1.2. Nicoll, who works in tourist destinations.**   Nicoll, from the DR, would travel for a few weeks at a time. She had a long history of visiting tourist destinations (e.g., Sosúa) outside of the capital Santo Domingo for work. She was familiar with the area and always stayed at the same hotel. She had regular clients and found new clients at the beach during the day or at dance clubs at night. She said it was easy to find clients there because these areas were known for sex work. Men went there with the intention of paying for sex. She also talked about the effects of police on her work when traveling to Puerto Plata, a beach town on the Atlantic north coast. She said, "Some businesses have closed, and the police have been bothering us [FSW] a lot. There is always abuse but now it is worse because the businesses have shut down and the women have to go on the street, and they jail most of them [. . .] That's why sometimes I do not go there, and I go to other towns where things are calm." She was the main caretaker for her granddaughters, who she referred to as her daughters. She viewed her mobility and that she had to leave her family as a hard battle that affected her health. Her main reason for being mobile was economic. "I travel because of my economic situation; my objective is to get money," she said. She would receive money and other types of resources from clients, especially from foreign men with whom she would sometimes spend two or three nights. Many tourist clients came back to the DR repeatedly to see her and became regular clients.

## 3.2. Characteristics of FSW's mobility in Tanzania and the DR

These two narratives illustrate how spatial, temporal, and social factors intersect to influence FSW's mobility in Tanzania and the DR. In Tanzania, mobility included travel to visit family, local mobility for sex work, and seasonal travel to specific destinations for sex work. In the DR, FSW traveled locally as well as outside of the capital, to the countryside, or to tourist areas, often for the purposes of sex work. Travel was common for many FSW, with some variation by destination. Additional social aspects of mobility–including economic need or aspects of FSW's risk environments–were critical to the context surrounding FSW's travel in both countries. Below, we describe these nuances and explore similarities and differences across settings.

**3.2.1. Spatial and temporal aspects of mobility.**   Figs 1 and 2 show participants' mobility in Tanzania and the DR. Participants traveled to local and more distant destinations during specific seasons, routinely, or ad hoc. Destinations were numerous and routes complicated, such as one participant (Shani) from Tanzania who traveled from Mtera and Dodoma (both north of Iringa) to Mbeya (southwest of Iringa) and Morogoro (east of Iringa) before returning to Iringa (Fig 1).

FSW described complex mobility narratives over different places and periods of time. For some, like Anisa in Tanzania, travel was common. "You know that we travel a lot" she said. FSW who were "shifting frequently," as Mary from Tanzania said, were contrasted with more permanent residents. Other travel, however, was less frequent or ad hoc. Lengths of trips varied by location and spanned from a day or two to weeks to even months, with trips to more distant destinations (e.g., Dar-es-Salaam and Zanzibar in Tanzania or tourist destinations in the DR) less frequent and lasting longer. There were also differences by FSW. Travel to some locations (i.e., Dodoma in Tanzania) lasted a week for one FSW and multiple months for another.

*Local mobility*: Local mobility included trips to visit family, to nearby health facilities, or for work. In Tanzania, travel to visit family was common. People visited their parents' home or traveled for ceremonies (e.g., weddings or funerals), often to places they were "from." Similar examples of trips to visit friends and family or vacations also emerged in the DR. In Tanzania in particular, some participants' family members and partners were also mobile, reflecting a broader context where mobility among family, community members, and clients as well as FSW was common.

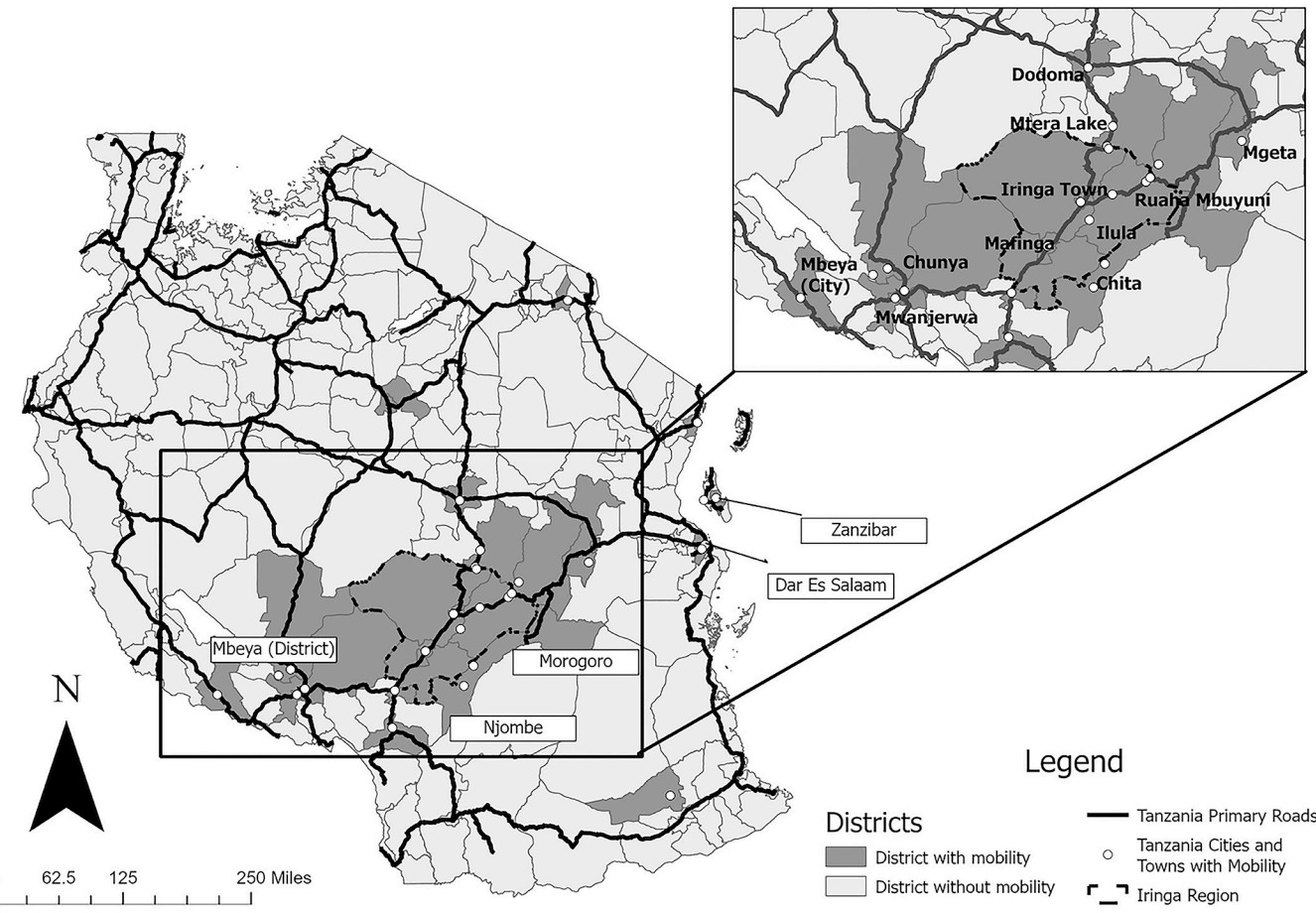

**Fig 1. Districts, cities, and/or towns in Tanzania named in participants' descriptions of mobility.** Maps were created using ArcGIS using basemap shapefiles from the UN Humanitarian Data Exchange, which is covered under the Creative Commons Attribution 4.0 International (CC BY 4.0). Specific basemap shape files were the United Republic of Tanzania—Subnational Administrative Boundaries [48] and the HOTOSM Tanzania Roads (OpenStreetMap Export) [49].

In both settings, FSW described clients as "passing" or "traveling here and there" and paying for sex in different locations. Women in Tanzania and the DR described how they would travel locally to meet clients or meet clients when away from home. They also met clients while walking, on public transportation, in a park, or in a hotel. This was facilitated by phone or apps that identified how far away someone was.

***Specific seasonal destinations*:** Women in both settings traveled to distant destinations (Figs 1 and 2). Locations included larger nearby cities and attractions such as tourist destinations, mining areas, or fishing communities, depending on the season. Harvest seasons led women in both settings to travel to specific places for sex work because the harvest meant that people had money to pay for sex. In Tanzania, participants traveled north to Dodoma for grape season or northeast to Morogoro for sesame and rice harvesting. In the DR, travel to the countryside coincided with the cacao harvest. In Tanzania, travel depended on whether it was the rainy season, as rain prevented mining and restricted travel to places like Dodoma. Peak fishing season and active mining seasons also influenced destinations.

*We are after fishermen's money. . . That is when fishing season is at its peak in Mtera [north of Iringa along the Mtera Reservoir], [and so] the fishermen have money. So, since I know all*

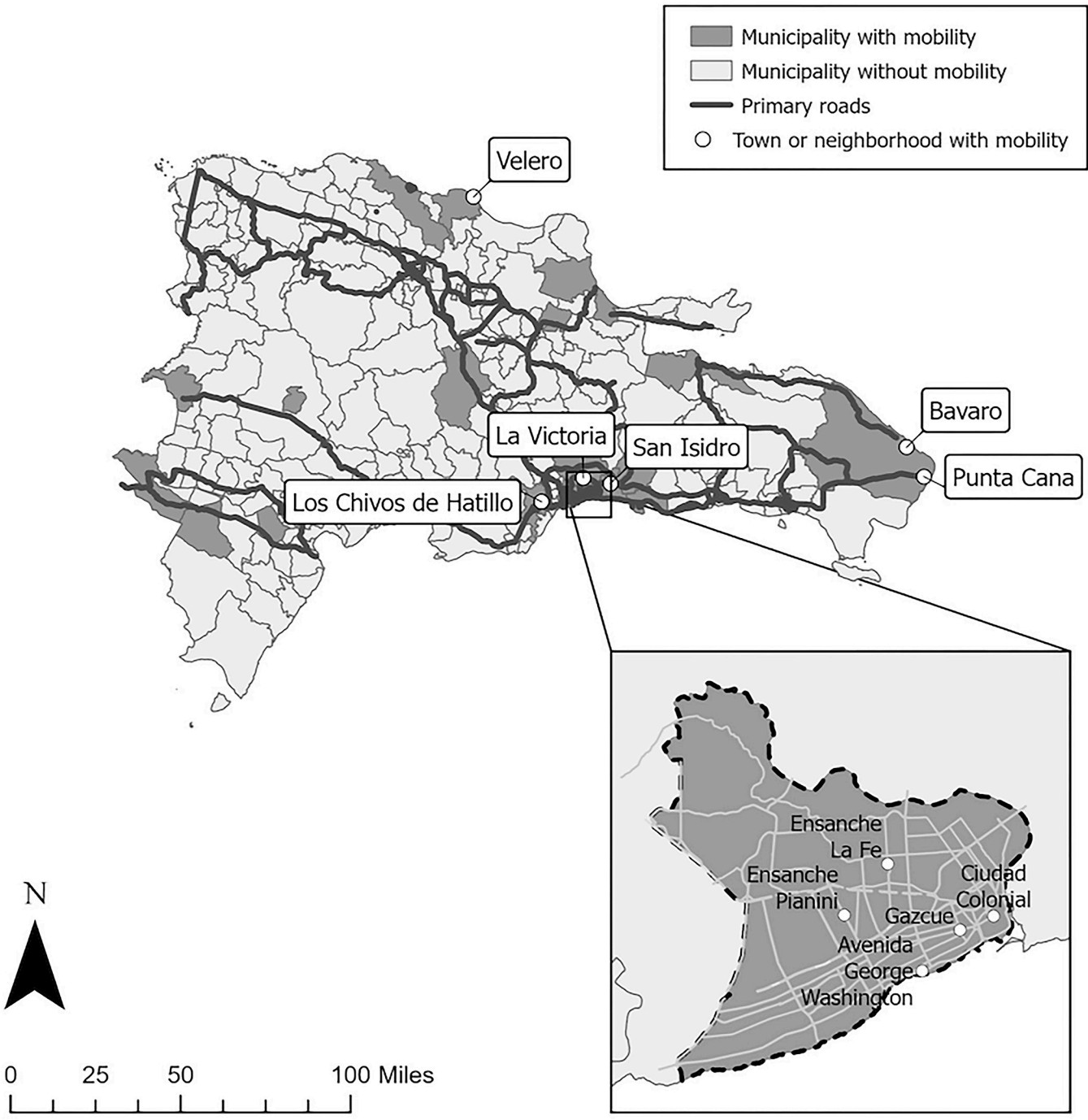

**Fig 2. Municipalities, towns, and/or neighborhoods in the Dominican Republic named in participants' descriptions of mobility.** Municipalities, towns, and/or neighborhoods in the Dominican Republic named in participants' descriptions of mobility. Maps were created using ArcGIS using basemap shapefiles from the UN Humanitarian Data Exchange, which is covered under the Creative Commons Attribution 4.0 International (CC BY 4.0). Specific basemap shape files were the Subnational Administrative Boundaries by Oficina national de Estadistica, OCHA Field Information Services Section [50] and the Dominican Republic Roads (OpenStreetMap Export) by Humanitarian OpenStreetMap Team (HOT) [51].

*the seasons, I know right now when I go to Mtera I will make money. So, I go there pretending to be a bar worker, but our main job is to trade sex for money [. . .] In Mbeya [southwest of Iringa] there is a place with a mine. The place is called Chunya. I usually know that in this particular season, minerals are found, [and so] most of the miners are found in that location. It is like the fishermen. (Shani, Tanzania)*

In the DR, trips were often taken to tourist destinations. Money could be earned during tourist season and low season.

*I have been traveling to Sosúa [beach town on the DR's Atlantic north coast] for five years [. . .] There are many foreign tourists there that I can meet. Even in low season when things are slow, there are always at least a few clients who go on vacation [. . .] I also have my old clients who always call me and keep in touch with me. (Nicoll, DR)*

**3.2.2. Social aspects of mobility.** Economic necessity, risk, the search for anonymity and familiarity, and social influence from fellow workers and clients emerged as key social factors influencing FSW's mobility experiences.

*Economic drivers*: Financial needs motivated participants to travel for work and leave one place for another. As Mary from Tanzania said, "[I travel for. . .] my basic needs so that I can get money for my needs to pay into *VICOBA* [village community banking], rent, to buy food, and to buy clothes for my children and my family" (Mary, Tanzania). The same was true in the DR. If earnings in one location did not meet expectations or were not sufficient for basic needs, participants would leave for another. Participants like Camila traveled to find clients to fulfill their economic needs, and locations were selected based on to clients' availability.

*I go to different local towns. I travel to El Cibao [region in the northern part of the country] to find Dominican clients there. If there aren't clients there, then I move on. I go to Bonao [center of the country, northwest of Santo Domingo]. Many men are traveling through Bonao, and even those from Bonao would stop there because they know they could buy sex there. (Camila, DR)*

*Response to risk*: Mobility was both a consequence of and a response to risk. Police and the risk of arrest were mentioned in both countries, with direct effects on the economic benefits of mobility, and led participants to avoid specific locations and opt for "calmer" places.

*She (a fellow FSW) would call me to let me know, "Do not come tomorrow. It is too 'caliente' (dangerous) here. The police are cracking down and arresting women." I would then change my plans and not travel at that moment. (Nicoll, DR)*

Other risks included potential harassment, such as the unwanted disclosure of one's HIV status, or violence. In one story, Anisa from Tanzania explained how a fellow worker and friend had shown other workers her HIV medication, which affected her work. "When I found that [out] I had to leave that place to go to another place," she said. Others described similar harassment, including being called a witch or told that they were killing other people's children. FSW in the DR also described the risk of verbal or physical violence, which led them to travel to a new location, opt for more rural areas, or reduce travel.

*Anonymity and familiarity*: Mobility for sex work was motivated by anonymity and familiarity. FSW perceived familiarity in both a positive and a negative light. Some, for instance, traveled to places where they did not have that familiarity. Participants explained a desire for

anonymity based on stigma related to sex work and their HIV status. Some expressed desires not to want to work at "home," where family or acquaintances live. Yari, from the DR, noted:

*I cannot work close to home. My granddaughters know I work in 'negocios' (businesses) as a waitress, and I travel to work in the 'negocios,' but they do not directly know that I go with clients to have sex. Even my eldest granddaughter does not know that I dedicate myself to that type of work. (Yari, DR)*

For Shani in Tanzania, the desire for anonymity meant that she did not stay in one place for long.

*Yes, we usually never stay in one area for a long time, it can be for just a week, just for a few days, when we are still considered as new to the place. We don't stay to the point of being familiar. It can just be two weeks. We take what we are after and move to another place. So, you can find that bar workers do move a lot. (Shani, Tanzania)*

Travel for work was also a consequence of FSW's desire to work where people did not know their HIV status. This search for anonymity was an economic decision for participants in both countries. FSW explained that regular customers did not pay as well as new customers, and once clients learned someone was living with HIV, their profits would shrink.

In contrast, FSW in both countries also described familiarity with a specific location or context as a benefit. Familiarity meant that one had connections in a place–often with people selling sex or business owners–or that one had gone to the same place repeatedly. As Nicoll in the DR said, familiarity with the place, the people, and the system contributed to perceptions that sex work would be less risky.

*I always travel to the same place. I already know the place, I know the system, you know, I have a place to stay, and I already know the people that stay in that hotel. I don't like to be risking too much, you know, traveling. . . If you have to explore too much to find clients it is too risky sometimes, but I like to play it safe. (Nicoll, DR)*

***Social relationships:*** Participants' relationships with friends, contacts, and clients varied across locations and influenced participants' mobility experiences. Women described how they would reach out to fellow workers and gather information before traveling.

*I ask her, "Hey, what is the situation like?" ["is there money?"] If they say it is fresh [alternatively translated as "great" or "awesome"], then I go. . . because you also see how difficult the situation is. (Anisa, Tanzania)*

FSW traveled together and built relationships with other mobile FSW as Anisa explained:

*After staying with her for two months, I left her there and came back to Iringa. After coming back to Iringa, I went again to Dodoma, although I didn't go to the place where she was staying. [. . .] She then went to Mozambique, [and] when she left Mozambique she started looking for me again, and we lived well together. She gave me something when I had nothing, and I gave her [something] when she had nothing. So we love each other. (Anisa, Tanzania)*

Clients also drove FSW's mobility in both countries. As Mary from Tanzania said, "Sometimes I get a client who drives and asks me to go with him. Mostly I go for the clients." Such

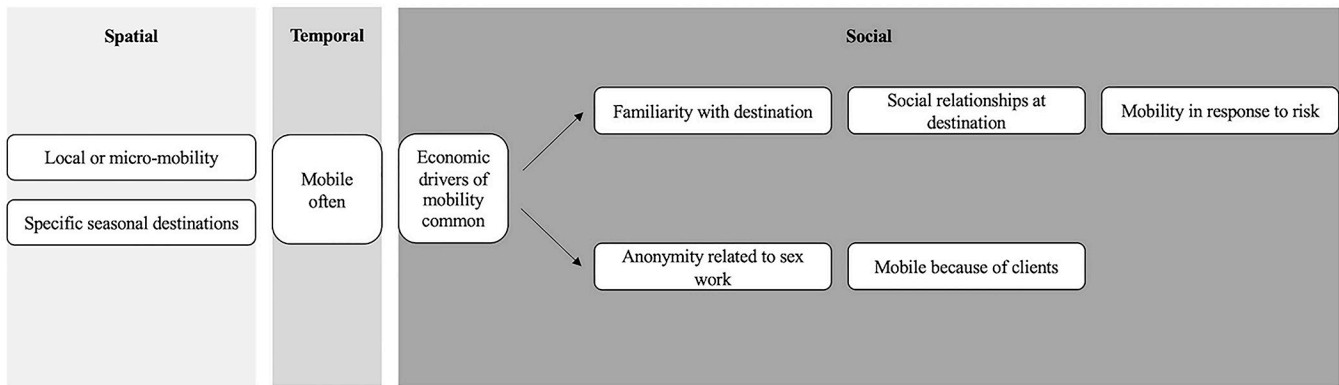

**Fig 3. Emergent mobility typologies in Tanzania and the Dominican Republic.**

trips were often unplanned, preventing women from taking HIV medications on time. Mary would, however, communicate her plans to other workers in case issues arose during the trip. Clients, as the ones paying, asserted their power over FSW to make travel decisions, particularly regarding destination and duration of travel. Elianny, a participant from the DR, said:

> *Sometimes the client wants me to stay longer. So, I stay more days than I had anticipated. The client decides and I follow what he wants because I need to keep him happy, and he pays me to stay extra nights. Sometimes it is hard for me because I do not have my medications because I did not plan to stay longer. (Elianny, DR)*

**3.2.3. Emergent mobility typologies in Tanzania and the DR.** Fig 3 shows mobility typologies that emerged in both countries. These typologies reflect patterns of spatial, temporal, and social factors described in previous sections that characterized FSW's mobility experiences and provide a visual representation of the narratives in sections 3.1.1, 3.1.2, and S1 Table.

In both settings, proximity of destination–either local mobility or more distant, specific seasonal destinations–was a key factor that differentiated FSW's travel. Among data analyzed here, FSW were often mobile, and economic drivers of mobility were common.

Two distinct typologies emerged in both countries, regardless of destination, frequency of travel, or economic motivations. For some FSW, the importance of familiarity with one's destination coincided with having social relationships at destination and mobility in response to risk. For other FSW, a desire for anonymity related to sex work intersected with traveling because of clients. These typologies consistently emerged in FSW's narratives but did not capture the full complexity of all mobility narratives.

## 4. Discussion

Across interviews with FSW living with HIV in Tanzania and the DR, participants illuminated spatial, temporal, and social factors that influenced mobility experiences. Destinations varied from local mobility to specific, seasonal locations. Travel included shorter and longer trips. FSW cited economic reasons for mobility and travel in response to risk, in search of anonymity, or motivated by familiarity with a particular location. Participants' social relationships played a critical role in their mobility, especially when traveling to familiar places.

FSW's stories support previous research in Tanzania demonstrating substantial travel within and outside the Iringa region [2], high levels of daily activity mobility among FSW living with HIV in the DR [4], and the influence of tourist destinations on sex work within the DR [10, 38]. Furthermore, they reinforce these FSW's engagement in what Hugo (2009) described as circular migration, or "a pattern of coming and going between a 'home' place and a destination place" [52]. Previous research with FSW in diverse settings, from Switzerland [53] to Mexico [54], has emphasized FSW's engagement in circular migration, highlighting the role repeated mobility plays in FSW's navigation of risk environments and sex work establishments [53]. Such work has highlighted the significance of circular migration for HIV transmission [54], yet further exploration of its impact on the health outcomes of FSW living with HIV is needed.

This study identified several motivations for mobility, including financial needs, visits to see family, risk (e.g., in response to police, harassment, or violence), and the desire for familiarity or anonymity. Many of these reasons have been cited by previous research [1, 11, 28, 30–32]. This qualitative investigation moves beyond spatial and temporal factors or reasons for mobility to understand the larger social context of travel, including the connection between mobility and one's social relationships. We identified emergent typologies (Fig 3) suggesting the potential co-occurrence of distinct social factors, such as 1) familiarity, social relationships, and risk or 2) anonymity and client-influence mobility, among FSW who traveled locally or to more distant seasonal destinations. These findings could inform the development of future quantitative typologies of FSW living with HIV.

Common across interviews was the influence of financial insecurity on mobility. Economic factors are a common motivation for mobility and migration, both among FSW [1] and non-FSW globally [55]. There is an urgent need to address the infrastructural determinants of health [56] that influence such mobility. Increasing access to microfinance, microcredit, savings groups, or other economic empowerment interventions is one approach used in Tanzania and other settings [57–59]. However, evidence from Tanzania [33] has highlighted suspicion surrounding mobile FSW's participation in savings groups, due to them dropping out or not fulfilling mutual financial obligations to the group. If mobility disrupts social relationships and cohesion [16, 34], it may also affect the trust necessary for the function of such savings groups [60]. Future community-based economic empowerment interventions could identify inclusive ways to engage mobile FSW sustainably. In other settings where gender inequitable norms restrict women's mobility and ability to participate in savings groups, digital technologies have been employed to address mobility-related barriers to participation in savings groups [61]. Similar approaches could be tested to facilitate engagement in VICOBA and other savings groups to address financial insecurity.

Social relationships played a key role in FSW's mobility experiences. These relationships linked mobile FSW across space and time, demonstrating how their experience of a place is influenced by the relationships and power dynamics that exist there and elsewhere [25, 26]. As Green (2019) wrote, "Daily life involves a density of multiple conditions, structures, regulations, environments, relations and separations that generate many versions of what it means to be somewhere in particular" [62]. Researchers have emphasized the potential benefits of social cohesion for FSW [63, 64]. However, interventions to foster social cohesion are often place-based, rather than focused on the needs of mobile populations. There is an opportunity for interventions–using digital technologies or other approaches–to amplify FSW's existing relationships to build social cohesion across space and increase access to social capital when mobile.

Emergent across interviews was the interplay between anonymity and familiarity; between traveling in search of anonymity about one's HIV status or not wanting to engage in sex work

"at home" and traveling in search of familiarity with place or people in that location. The desire for anonymity may link FSW's mobility with intersectional stigma [65] related to their engagement in sex work or HIV status. Future research could investigate the interrelationship between mobility and intersectional stigma to identify challenges and new avenues for community empowerment programs.

These findings reflected a first, exploratory step in identifying typologies of FSW mobility (Fig 3). Future research could draw on these themes to inform quantitative and qualitative research and interventions focused on FSW mobility. Our findings suggest priority questions for quantitative surveys, with the emergent typologies having the potential to inform hypotheses for latent class analyses and other audience segmentation techniques to use to understand the typologies that are more or less common and how they are related to healthcare utilization. This work can inform the design and implementation of tailored, community-based HIV care and treatment interventions that meet the distinct needs of FSW living with HIV with diverse stories of mobility [66]. Those who engage in short-term, local mobility may require, for instance, different adaptive community-based solutions for improving ART adherence or engagement in HIV care at the local level as compared to those who travel seasonally to more distant tourist destinations or mines [27], who may require multi-sited interventions or digital technologies to address barriers to accessing HIV services. Similarly, those who travel in search of anonymity may have unique needs for accessing information, social support, HIV medications, or HIV care than do those who travel routinely to the same familiar places. Around the world, mobility is increasingly a component of people's daily lives and livelihoods, be it in response to economic circumstances, climate change, emergency, or other push and pull factors [67]. Researchers can learn from the lived experiences of FSW to inform how to improve HIV care and treatment interventions moving forward for those who are mobile.

### 4.1 Strengths and limitations

There are several considerations for interpreting the findings of this study. First, qualitative findings are not intended to be representative but instead highlight similarities and differences in spatial, temporal, and social factors influencing FSW's mobility experiences in Tanzania and the DR. We explored the particularities of individual stories and identified emergent themes. Consistency in themes across interviews in two settings suggest that they may be transferable to other contexts. Further qualitative and quantitative research could assess the extent to which they reflect distinct typologies of mobility among FSW living with HIV and their theoretical transferability to other contexts.

Second, these findings reflect the perspectives of individuals with recent mobility. If mobility is a process that takes place over time and space [21–23], women's stories may differ by their position in that process. For example, participants' reflections on what influenced decisions to travel could be influenced by when or where or with whom they most recently traveled and where they are in that mobility (e.g., before, during, or after travel).

Third, this study is strengthened by the rigor of the analytical approach employed. The utilization of qualitative and quantitative data to select stories for analysis enabled the research team to gather a more robust picture of participants' mobility experiences. Thematic analytical methods allowed us to identify themes across participants' experiences that then informed the narrative analysis conducted. The narrative analysis prioritized participants' complete stories. By integrating categorizing and connecting analytical approaches, we demonstrated how cross-cutting themes emerged and intersected in unique ways across participants and settings.

Finally, while this study is strengthened by data collection at multiple time points, the realities of life for FSW in Tanzania and the DR may have posed challenges to retention. Some

participants in the parent study were only interviewed once, while questions focused on mobility were asked during the second, follow-up interview. As mobility is a common reason for loss to follow up for FSW [15], it is possible that those who did not participate in both rounds of data collection were lost to follow up due to mobility. Inclusion of questions on mobility at multiple time points, and using mobile technologies to facilitate follow up, could improve retention and facilitate longitudinal analyses.

## 5. Conclusions

This study illustrated how mobility is understood and implicated in the lived experiences of FSW living with HIV in Tanzania and the DR. Mobility is increasingly recognized as influential in FSW's access to and utilization of HIV-related care and treatment services, with direct implications for health and well-being over time. However, current quantitative measures of mobility among this population are limited, with nuances abstracted to categorize anyone with mobility experience, regardless of spatial, temporal, or social characteristics, as mobile. This study illustrates that mobility is experienced in distinct ways that can lead FSW living with HIV to have unique needs in terms of information or HIV care and treatment. Future research should explore how HIV care and treatment interventions with FSW could be tailored to ensure that resources are efficiently applied to meet FSW's needs in each setting.

## Supporting information

**S1 Checklist. Inclusivity in global research.**
(DOCX)

**S1 Table. Supplemental mobility narratives of FSW in Tanzania and the Dominican Republic.**
(DOCX)

## Author Contributions

**Conceptualization:** Zoé M. Hendrickson, Maria De Jesus, Clare Barrington, Caitlin E. Kennedy, Yeycy Donastorg, Martha Perez, Hoisex Gomez, Jessie Mbwambo, Samuel Likindikoki, Deanna L. Kerrigan.

**Formal analysis:** Zoé M. Hendrickson, Maria De Jesus, Laura Nicole Sisson, Mudia Uzzi.

**Funding acquisition:** Zoé M. Hendrickson, Clare Barrington, Yeycy Donastorg, Martha Perez, Hoisex Gomez, Jessie Mbwambo, Samuel Likindikoki, Deanna L. Kerrigan.

**Investigation:** Clare Barrington, Yeycy Donastorg, Martha Perez, Hoisex Gomez, Jessie Mbwambo, Samuel Likindikoki, Deanna L. Kerrigan.

**Methodology:** Zoé M. Hendrickson, Maria De Jesus, Clare Barrington, S. Wilson Cole, Caitlin E. Kennedy, Yeycy Donastorg, Martha Perez, Hoisex Gomez, Jessie Mbwambo, Samuel Likindikoki, Deanna L. Kerrigan.

**Project administration:** S. Wilson Cole, Yeycy Donastorg, Martha Perez, Hoisex Gomez, Jessie Mbwambo, Samuel Likindikoki, Deanna L. Kerrigan.

**Resources:** Deanna L. Kerrigan.

**Software:** Zoé M. Hendrickson.

**Supervision:** Caitlin E. Kennedy, Deanna L. Kerrigan.

**Visualization:** Laura Nicole Sisson, Mudia Uzzi.

**Writing – original draft:** Zoé M. Hendrickson.

**Writing – review & editing:** Zoé M. Hendrickson, Maria De Jesus, Clare Barrington, S. Wilson Cole, Caitlin E. Kennedy, Laura Nicole Sisson, Mudia Uzzi, Yeycy Donastorg, Martha Perez, Hoisex Gomez, Jessie Mbwambo, Samuel Likindikoki, Deanna L. Kerrigan.

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
